# Disentangled Embedding through Style and Mutual Information for Domain Generalization

**Noaman Mehmood**                                                       *noaman@udel.edu*
*Department of Electrical and Computer Engineering*
*University of Delaware*

**Kenneth Barner**                                                       *barner@udel.edu*
*Department of Electrical and Computer Engineering*
*University of Delaware*

**Reviewed on OpenReview:** *https://openreview.net/forum?id=552tedTByb*

## Abstract

Deep neural networks often experience performance degradation when faced with distributional shifts between training and testing data, a challenge referred to as domain shift. Domain Generalization (DG) addresses this issue by training models on multiple source domains, enabling the development of invariant representations that generalize to unseen distributions. Although existing DG methods have achieved success by minimizing variations across source domains within a shared feature space, recent advances inspired by representation disentanglement have demonstrated improved performance by separating latent features into domain-specific and domain-invariant components. We propose two novel frameworks: Disentangled Embedding through Mutual Information (DETMI) and Disentangled Embedding through Style Information (DETSI). DETMI enforces disentanglement by employing a mutual information estimator, minimizing the mutual dependence between domain-agnostic and domain-specific embeddings. DETSI, on the other hand, achieves disentanglement through style extraction and perturbation, facilitating the learning of domain-invariant and domain-specific representations. Extensive experiments on the PACS, Office-Home, VLCS and TerraIncognita datasets show that both frameworks outperform several state-of-the-art DG techniques.

## 1 Introduction

Deep neural networks excel at learning discriminative features and achieve outstanding performance in computer vision tasks. However, their performance deteriorates significantly when tested on data with a distribution different from the training data Ben-David et al. (2010). For instance, a network trained on images from an urban environment performs poorly when applied to rural environments. This performance drop results from the domain shift Quiñonero-Candela et al. (2022), which refers to the difference between the source and target data distributions. To address this issue, Domain Generalization (DG) was introduced in 2011 Blanchard et al. (2011). DG proposes training a model on multiple relevant source domains to develop invariant representations that generalize to new distributions not present during training.

During the past 12 years, numerous approaches have been proposed to address the challenge of DG. Among these, adversarial training has been widely explored to encourage domain-invariant feature learning in different domains Li et al. (2018b;c); Sinha et al. (2017). Other strategies have employed meta-learning Li et al. (2018a); Balaji et al. (2018); Zhao et al. (2021); Finn et al. (2017), which enables models to learn to generalize by simulating domain shifts during training. Ensemble learning Rame et al. (2022) is another popular method in domain generalization, where multiple models are trained with diverse data views, and their predictions are aggregated to improve the robustness against domain changes.

Data augmentation techniques have also emerged as promising solutions, in which synthetic data is generated to expose the model to a wider range of variations, helping it generalize to unseen domains Mehmood & Barner (2024); Volpi et al. (2018); Zhou et al. (2020a;b). Furthermore, some methods focus on learning domain-invariant features, such as shape representations, by minimizing the divergence between latent representations between domains, thereby improving generalization to new environments Zhou et al. (2022); Lu et al. (2022). These approaches represent key advances in DG, each contributing unique mechanisms to address the distributional shifts that lead to domain shifts.

Most of these methods discard domain-specific information (*e.g*, background, style) to minimize the divergence between the embeddings from different source domains. Recently, inspired by disentangled representation learning, the authors of POEM Jo & Yoon (2023) trained a model to learn domain-invariant and domain-specific information separately. In addition, to remove redundant information between these two feature spaces, cosine similarity was minimized, further enhancing disentanglement.

Inspired by disentangled embedding learning POEM Bui et al. (2021); Jo & Yoon (2023); Yu et al. (2024) for domain generalization (DG), we introduce two novel frameworks designed to effectively learn domain-agnostic embeddings by leveraging domain-specific information. These frameworks enable the model to explicitly factorize the representation space, allowing for more targeted learning of invariant features critical for out-of-distribution generalization. By decoupling the domain-specific noise and task-relevant signal, we aim to enhance robustness and transferability across unseen domains.

In our first proposed framework, named DETMI, we leverage domain-specific knowledge to decouple domain-invariant representations. To this end, we introduce a domain-specific encoder and classifier alongside a task-specific encoder and classifier. To improve disentanglement, POEM Jo & Yoon (2023) employs cosine similarity to promote linear independence between domain-specific and domain-agnostic feature spaces. However, cosine similarity does not account for higher-order dependencies, which can result in residual information leakage between the two spaces and compromise the effectiveness of the disentanglement. To overcome this limitation, we propose minimizing the mutual information between the two feature spaces. This strategy reduces higher-order correlations and facilitates the learning of more robust and statistically independent representations, thereby enhancing generalization to unseen domains. The DETMI framework goes beyond geometric constraints by offering a principled mechanism for enforcing independence, thereby enabling deeper statistical disentanglement of the underlying factors of variation.

In our second approach, we introduce the DETSI framework, which also leverages domain-specific knowledge through a domain-specific encoder and classifier. Additionally, to improve disentanglement, it incorporates style information to separate domain-specific and domain-agnostic embeddings. Inspired by Huang & Belongie (2017), which establishes a correlation between the domain of an image and its style, we apply style perturbation to encourage the extraction of domain-agnostic embeddings. These embeddings prioritize high-level features, such as object structure and semantic content, over domain-dependent attributes. This approach reduces the model's reliance on superficial cues and promotes the learning of features that generalize across domain shifts. Meanwhile, for domain-specific embeddings, DETSI captures style-specific characteristics, including texture and artistic style, rather than low-level features like edges or pixel intensity. By explicitly modeling style-based attributes, DETSI effectively disentangles domain-specific components from domain-invariant ones and produces robust embeddings, thereby significantly enhancing generalization across diverse unseen domains.

We conducted extensive experiments on the PACS Li et al. (2017), Office-Home Venkateswara et al. (2017), VLCS Torralba & Efros (2011), and TerraIncognita Beery et al. (2018) benchmarks. The results demonstrate that the proposed approaches outperform several state-of-the-art (SOTA) methods addressing Domain Generalization.

The remainder of this paper is organized as follows. Section 2 reviews related work, summarizing key methods developed to address the challenges of domain generalization. Section 3 details the proposed methodologies, including the techniques used to tackle domain generalization. Section 4 describes the experimental setup, covering datasets, implementation details, and the results of the proposed methods. Section 5 presents ablation studies, providing deeper insights into the contributions of individual components. Finally, Section 6 concludes the paper by summarizing the findings and discussing their implications.

## 2  Related Work

Domain Generalization (DG)  Zhou et al. (2020a) is an approach that trains models using labeled data from multiple source domains to generalize to an unseen target domain effectively. This problem arises naturally in applications such as medical imaging, autonomous driving, and visual recognition, where the distribution of data in the real world may differ significantly from that in the training set, leading to poor generalization performance. Over the past decade, numerous techniques have been proposed to tackle domain generalization, including domain-invariant feature learning, which focuses on extracting representations that remain consistent across domains; meta-learning, which leverages learning-to-learn paradigms for better adaptability; data augmentation, which generates diverse training examples to improve model robustness; adversarial learning, which introduces adversarial objectives to align distributions across domains; and disentangled embedding learning, which separates domain-specific and domain-invariant factors to enhance generalization. Each of these approaches is discussed in detail in the following subsections.

### 2.1  Domain-Invariant Representation Learning

One of the most popular strategies in domain generalization is to learn domain-invariant representations, where features are shared across different domains and remain stable under distributional shifts. Early works in this area have leveraged feature alignment techniques to ensure that representations of different domains become indistinguishable in a learned latent space. For example, Muandet *et al.* Muandet et al. (2013) introduced Domain-Invariant Component Analysis (DICA), where they aimed to learn a feature transformation that removes domain-specific variations while retaining the information necessary for classification. Similarly, Ghifary *et al.* proposed Scatter Component Analysis (SCA) Ghifary et al. (2016), a method that projects data into a subspace where the variance between domains is minimized and the class separation is maximized.

Another approach uses distribution matching techniques like Maximum Mean Discrepancy (MMD) to align the feature distributions across domains. For example, Li *et al.* Li et al. (2018b) proposed the Domain-Adversarial Neural Network (DANN) framework, where an adversarial loss is used to align feature distributions between domains by confusing a domain classifier that attempts to distinguish between source domains.

### 2.2  Meta-Learning

Meta-learning, often called "learning to learn," has recently gained traction as a promising framework for domain generalization. Meta-learning approaches Li et al. (2018a); Zhao et al. (2021); Finn et al. (2017) aim to train a model to quickly adapt to new, unseen domains. In this method, data from source domains is divided into meta-train and meta-test segments, allowing the model to be trained specifically to excel on the meta-test data using the meta-train data, mimicking real-world applications where the model must adapt to completely new data.

Li *et al.* Li et al. (2018a) introduced a meta-learning framework for domain generalization in which the model is trained on multiple source domains to simulate the process of generalizing to new, unseen domains. Specifically, the model is trained in a meta-learning loop, where each training iteration mimics the domain generalization process by exposing the model to different domain shifts, helping the model learn robust features that generalize well to unseen domains.

Another key work is by Balaji *et al.* Balaji et al. (2018), who proposed MetaReg, where a meta-regularization term is learned to guide the model's parameters to be domain-agnostic. These approaches focus on teaching the model to adapt quickly to new tasks by simulating domain shifts during training. Meta-learning approaches have shown great promise in domain generalization due to their ability to simulate and adapt to new domain distributions in the training phase.

### 2.3    Adversarial Learning

Adversarial learning has also been widely used in domain generalization to reduce the gap between source and unseen target domains. The core idea is to learn indistinguishable feature representations across domains using adversarial training techniques.

In one of the foundational works, Li *et al.*Li et al. (2018b) proposed an adversarial autoencoder (AAE), where the encoder is adversarially trained to produce features that are domain-invariant, while a domain discriminator is trained to distinguish between features from different domains. The feature extractor tries to fool the domain discriminator, thus learning domain-invariant representations.

In another work, Li *et al.* Li et al. (2018c) train a conditional invariant adversarial network to learn domain-invariant representations by making the learned representations on different domains indistinguishable through adversarial training.

Adversarial approaches have proven to be effective in learning representations that are robust to domain shifts, though they often require careful tuning of the adversarial loss function.

### 2.4    Data Augmentation

In addition to improving DG, data augmentation techniques Zhou et al. (2020a;b); Li et al. (2023) introduce more variety to training data by augmenting existing data pairs $(x, y)$, where $x$ represents the input and $y$ the corresponding label. These techniques generate transformed pairs $(A(x), y)$, where $A(\cdot)$ is a transformation that preserves the original label. This process helps to prepare the model to handle the diverse conditions encountered in the source domains.

Volpi *et al.* Volpi et al. (2018) proposed an augmentation strategy based on adversarial perturbations, where synthetic examples are generated by perturbing the original data to mimic potential unseen domains. The model can better handle domain shifts at test time by training on these perturbed examples.

Shankar *et al.* Shankar et al. (2018) introduced CrossGrad, an approach that uses the gradient of the domain classifier to perturb input examples. This ensures that the generated examples lie closer to the decision boundary of the domain classifier, forcing the model to learn domain-invariant features.

Zhou *et al.* Zhou et al. (2020b) proposed a data augmentation approach that utilizes a data generator to synthesize samples from pseudo-novel domains, effectively expanding the source domain with artificially generated data. By creating these domain variations, the model is exposed to a wider range of potential domain shifts, enhancing its ability to generalize to unseen target domains.

In another work, Zhou *et al.* Zhou et al. (2020a) developed a Deep Domain-Adversarial Image Generation (DDAIG) network to generate more synthetic data using adversarial training, increasing domain diversity, and improving the generalization capabilities of the model. Other techniques like Domain Randomization (DR)  Tobin et al. (2017) and image transformations, which adjust visual features such as color, texture, and lighting, add further robustness.

Data augmentation methods have been particularly effective when the goal is to simulate diverse, unseen domains during training. However, a challenge with these methods is ensuring that the generated augmentations accurately reflect the domain shifts encountered in practice.

### 2.5    Disentangled Embeddings Learning

Inspired by representation learning, recent advances in disentangled embedding learning for domain generalization have increasingly focused on partitioning the latent space into domain-invariant and domain-specific embeddings. Researchers, including  Bui et al. (2021) and  Jo & Yoon (2023), minimize metrics such as covariance or cosine similarity between these spaces to enforce greater independence. Using a similar intuition, Yu et al. (2024), minimize KL divergence loss to learn domain-agnostic and domain-specific embeddings through a single encoder. This separation enhances the model's generalization ability across diverse and unseen domains.

Our proposed frameworks tackle DG by disentangling latent representations into domain-specific and domain-invariant components. We leverage domain-specific information to aid in the learning of domain-invariant semantic features, ensuring a more robust representation. We leverage mutual information and style information in DETMI and DETSI, respectively, to effectively capture both domain-related and domain-agnostic embeddings. Through this disentanglement, our proposed approaches improve the model's ability to learn well-separated representations, leading to improved generalization across diverse domains.

## 3 Proposed Method

The feature and label spaces are represented by $\mathcal{X} \subset \mathbb{R}^D$ and $\mathcal{Y} \subset \mathbb{R}$, respectively. A domain is represented by a joint distribution $P_{xy} \in \mathcal{P}_{\mathcal{X} \times \mathcal{Y}}$, where $\mathcal{P}_{\mathcal{X} \times \mathcal{Y}}$ denotes the set of joint probability distributions on $\mathcal{X} \times \mathcal{Y}$. In DG, we have access to $K$ similar but distinct source domains as $S^k = (x_i^{(k)}, y_i^{(k)})_{i=1}^{N^{(k)}}$, each associated with a joint distribution $P_{xy}^{(k)}$, where $(x_i^{(k)}, y_i^{(k)}) \sim P_{xy}^{(k)}$ and $N^{(k)}$ denote the total number of data points in a particular domain $k$. The goal of DG is to learn a model $f : \mathcal{X} \to \mathcal{Y}$ using data from the source domain so that it can generalize well to an unseen target domain $\tau$ having a joint distribution $P_{xy}^\tau$.

The next two subsections introduce frameworks for disentangling embeddings to improve domain generalization. The first subsection, Disentangled Embedding Through Mutual Information (DETMI), presents a framework that leverages mutual information between domain-specific and domain-invariant components to achieve learning of generalizable features. The second subsection, Disentangled Embedding Through Style Information (DETSI), proposes a complementary strategy that uses style information, such as feature statistics and Gram matrices, to isolate domain-specific and domain-agnostic features. This framework promotes the separation of domain-relevant components by focusing on content and style perturbations, further improving the model's ability to generalize across unseen domains.

### 3.1 Disentangled Embedding through Mutual Information (DETMI)

DG assumes the invariance between $\mathcal{X}$ and $\mathcal{Y}$. Existing methodologies primarily focus on acquiring invariant features from available source domains, anticipating that these invariant features can be extended to predict the target domains unseen during training. However, these approaches remain susceptible to erroneous predictions because of their inability to handle variations caused by data bias. Domain-specific features such as background, style variation, and location introduce data bias that adversely affects the prediction performance of a model. Drawing inspiration from disentangled embedding learning Bengio et al. (2013), we leverage these domain-specific cues to disentangle domain-agnostic representations and extract more generalizable features. By separating domain-specific information from domain-invariant components, our approach enhances the model's ability to generalize to unseen target domains.

To this end, we propose the Disentangled Embeddings Through Mutual Information (DETMI) framework illustrated in Fig. 1. In the framework, we train two encoders, $E_c$ and $E_d$, to learn category-related and domain-specific characteristics denoted as $Z_c$ and $Z_d$, respectively. We also employ two classifiers, $C$ and $\tilde{C}$, to predict class labels and domain labels, respectively. The class and domain classifiers are trained using cross-entropy losses $\mathcal{L}_c$ and $\mathcal{L}_d$, defined as:

$$\mathcal{L}_c = -y_i^k \log \left( \sigma \left( C(E_c(x_i^k; \theta_c); \delta_c) \right) \right), \tag{1}$$

$$\mathcal{L}_d = -d_i^k \log \left( \sigma \left( \tilde{C}(E_d(x_i^k; \theta_d); \delta_d) \right) \right), \tag{2}$$

where $y_i^k$ and $d_i^k$ denote the class and domain labels, respectively. The parameters $\delta_c$, $\delta_d$, $\theta_c$, and $\theta_d$ correspond to the classifiers $C$, $\tilde{C}$, and the encoders $E_c$ and $E_d$, respectively. The function $\sigma$ denotes the sigmoid activation.

The optimization objective function for learning domain-agnostic feature space through the leverage of domain-specific information is:

$$\mathcal{L}_{dI} = \mathcal{L}_c + \lambda_1 \mathcal{L}_d. \tag{3}$$

where $\lambda_1$ is the weighting coefficient.

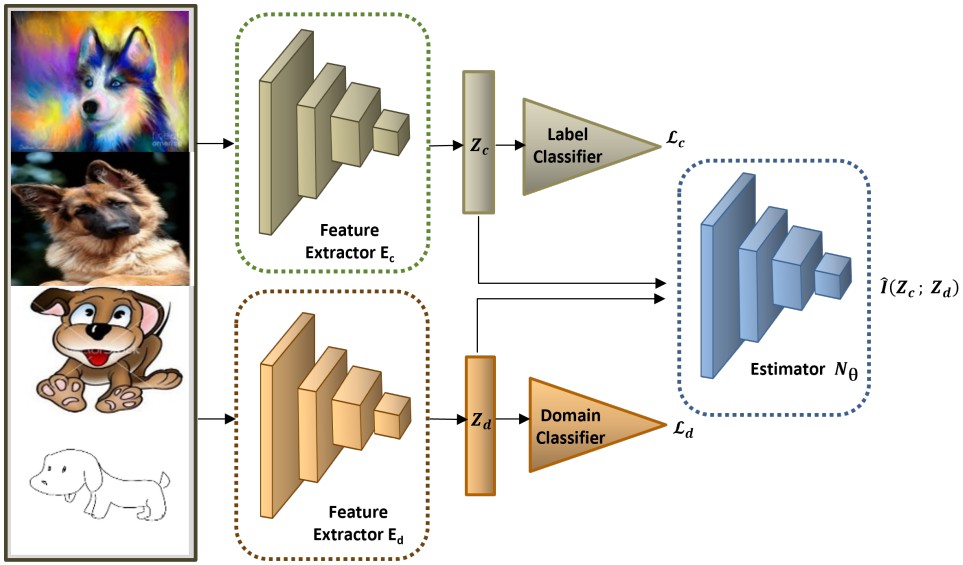

Figure 1: DETMI: Disentangled Embedding Through Mutual Information

Although we leverage domain-specific features to learn only domain-agnostic features, this constraint is insufficient to remove all domain-relevant information. A recent study Jo & Yoon (2023) tackled this challenge by employing cosine similarity to encourage geometric independence between these feature spaces, thus improving disentanglement. Our approach minimizes mutual information between $Z_c$ and $Z_d$ to enforce independence, addressing higher-order correlations beyond the mere spatial arrangement.

### 3.1.1 Independence through Mutual Information

Let $X$ and $Y$ be two random variables. Mutual information $I(X;Y)$ measures the statistical dependence between two variables.

**Theorem 1.** If $I(X;Y) = 0$, then $X$ and $Y$ are statistically independent.

**Proof.** The mutual information between two random variables $X$ and $Y$ is given by:

$$I(X;Y) = \iint P(x,y) \log \frac{P(x,y)}{P(x)P(y)} \, dx \, dy. \tag{4}$$

If $I(X;Y) = 0$, then:

$$\log \frac{P(x,y)}{P(x)P(y)} = 0 \Rightarrow P(x,y) = P(x)P(y). \tag{5}$$

In light of Theorem 1, a well-known result in information theory, reducing $I(X;Y)$ brings $P(x,y)$ closer to $P(x)P(y)$, promoting statistical independence. This facilitates the learning of semantically meaningful disentangled embeddings, enhancing the extraction of generalizable features for improved domain adaptation.

The mutual information between $Z_c$ and $Z_d$ is given by:

$$I(Z_c; Z_d) = \iint p(\tilde{z}, \hat{z}) \log \left( \frac{p(\tilde{z}, \hat{z})}{p(\tilde{z})p(\hat{z})} \right) d\tilde{z} \, d\hat{z} \tag{6}$$

where $p(\tilde{z}, \hat{z})$ is the joint probability density function of $Z_c$ and $Z_d$, and $p(\tilde{z})$ and $p(\hat{z})$ are the marginal probability density functions of $Z_c$ and $Z_d$, respectively. For high-dimensional variables, the calculation of the double integrals is quite complex.

MINE Belghazi et al. (2018), utilizes the Donsker-Varadhan representation of the Kullback-Leibler (KL) divergence to provide a lower bound on mutual information and optimize it using a neural network.

Using Donsker-Varadhan representation, the Kullback-Leibler (KL) divergence between two probability distributions $P$ and $Q$ is given by:

$$D_{\mathrm{KL}}(P||Q) = \sup_{T:\Omega\to\mathbb{R}} \mathbb{E}_P[T] - \log\mathbb{E}_Q[e^T]. \tag{7}$$

where the supremum is taken over all functions $T$ for which both expectations remain finite.

The mutual information $I(X;Y)$ between two random variables $X$ and $Y$ is defined using the Kullback-Leibler (KL) divergence as:

$$I(X;Y) = D_{\mathrm{KL}}(P(X,Y)||P(X)P(Y)), \tag{8}$$

where $P(X,Y)$ is the joint probability distribution of $X$ and $Y$. $P(X)P(Y)$ represents the product of marginal distributions.

Following MINE, we employ a neural network $N_\theta$ to estimate the lower bound of $I(Z_c; Z_d)$:

$$I(Z_c; Z_d) \geq \hat{I}(Z_c; Z_d) = \sup_{\theta} \mathbb{E}_{p(\tilde{z},\hat{z})}[N_\theta] - \log\mathbb{E}_{p(\tilde{z})\otimes p(\hat{z})}[e^{N_\theta}]. \tag{9}$$

The expectations in (9) are computed using the approach of MINE. Accordingly,

$$\hat{I}(Z_c; Z_d) = \frac{1}{n}\sum_{i=1}^{n} N(\tilde{z}_i, \hat{z}_i, \theta) - \log\left(\frac{1}{n}\sum_{i=1}^{n}\exp^{N(\tilde{z}_i, \tilde{\tilde{z}}_i, \theta)}\right), \tag{10}$$

where $(\tilde{z}, \hat{z})$ is obtained through the joint probability density function $p(\tilde{z}, \hat{z})$ and $\tilde{\tilde{z}}$ is obtained through the marginal distribution $p(\hat{z})$ by a random shuffle.

By minimizing $\hat{I}(Z_c; Z_d)$, we force the learned embeddings $Z_c$ and $Z_d$ to satisfy:

$$P(Z_c, Z_d) \approx P(Z_c)P(Z_d). \tag{11}$$

This prevents information leakage from $Z_d$ into $Z_c$, ensuring proper disentanglement. The final optimization objective function for learning domain-agnostic feature space through the leverage of domain-related information is:

$$\mathcal{L}_{dI} = \mathcal{L}_c + \lambda_1\mathcal{L}_d + \lambda_2\hat{I}(Z_c; Z_d). \tag{12}$$

where $\lambda_1$ and $\lambda_2$ are the weighting coefficients.

---

**Algorithm 1** Training procedure for DETMI

---

1: **Input:** $K$ domains data samples; Encoders $E_c, E_d$; Classifiers $C, \tilde{C}$; Mutual information estimator $N_\theta$; Batch size B
2: **Output:** Optimized Encoder $E_c$ and Classifier $C$
3: Using MINE, Update Mutual information estimator $N_\theta$ by maximizing 10 until convergence.
4: **for** $i = 1$: epochs **do**
5:     Sample a mini-batch $(x_i^{(k)}, y_i^{(k)})_{i=1}^{B} \in S^k$
6:     Compute the objective function
7:     $\mathcal{L}_{dI} = \mathcal{L}_c + \lambda_1\mathcal{L}_d + \lambda_2\hat{I}(Z_c; Z_d).$
8:     Update $E_c, E_d, C, \tilde{C}$
9: **end for**
10: **Return** Optimized Encoder $E_c$ and Classifier $C$

---

### 3.2 Disentangled Embedding through Style Information (DETSI)

We propose DETSI, an alternative approach for learning domain-agnostic embeddings by leveraging domain-specific information. DETSI leverages style information to learn domain-agnostic and domain-specific embeddings, denoted as $Z_c$ and $Z_d$. Building on the work of Huang & Belongie (2017), which establishes the close relationship between style and instance-level feature statistics, and recognizing the strong correlation between image style and visual domains, we employ style perturbation to encourage content-focused learning for domain-agnostic embeddings. Additionally, we extract style features to enhance the learning of domain-specific information, enabling a more effective separation of domain-specific and domain-invariant components.

#### 3.2.1 Preliminaries

Huang & Belongie (2017) demonstrated that CNN convolutional features maps statistics i.e. channel-wise mean and variance, effectively characterize image style. Building on this insight, Ulyanov et al. (2016) proposed Instance Normalization (IN) to normalize these style statistics and mitigate style variations in style transfer models.

For a given input image $x$, its feature maps are represented as $f_x \in \mathbb{R}^{C \times H \times W}$, where $C$ denotes the number of channels, and $H$ and $W$ correspond to the spatial dimensions. The formulation of Instance Normalization (IN) is expressed as:

$$\text{IN}(f_x) = \gamma \frac{f_x - \mu_f}{\sigma_f} + \beta, \tag{13}$$

where $\gamma, \beta \in \mathbb{R}^C$ correspond to learnable affine transformation parameters, and $\mu_f, \sigma_f \in \mathbb{R}^C$ correspond to the channel-wise mean and standard deviation.

$$\mu_f = \frac{1}{HW} \sum_{h=1}^{H} \sum_{w=1}^{W} f_{c,h,w}, \tag{14}$$

and

$$\sigma_f = \sqrt{\frac{1}{HW} \sum_{h=1}^{H} \sum_{w=1}^{W} (f_{c,h,w} - \mu_f)^2 + \epsilon}, \tag{15}$$

where a small constant $\epsilon$ is added to avoid numerical instability.

Furthermore, leveraging these style statistics, Huang & Belongie (2017) introduced Adaptive Instance Normalization (AdaIN), which transfers an image's style to a target style by replacing the affine parameters with corresponding style-specific statistics $(\mu_s, \sigma_s)$. AdaIN is defined as:

$$\text{AdaIN}(f_x, s) = \sigma_s \frac{f_x - \mu_f}{\sigma_f} + \mu_s. \tag{16}$$

In this paper, we introduce perturbations to the channel-wise mean and standard deviation of each feature map to introduce style randomization. Furthermore, we utilize AdaIN to substitute the original style information with randomly generated style statistics, enhancing style invariance in our feature representations.

#### 3.2.2 Content-focused Learning

We achieve content-focused learning by introducing style perturbation, which alters the feature statistics at the instance level of the training images. These feature statistics, closely tied to style, are perturbed to encourage the model to focus on content rather than stylistic variations. The process is formally defined by:

$$\mu_{new} = \lambda \mu(f_c^{(l)}) + (1 - \lambda)\mu(\tilde{f}_c^{(l)}), \tag{17}$$

and

$$\sigma_{new} = \lambda\sigma(f_c^{(l)}) + (1 - \lambda)\sigma(\tilde{f}_c^{(l)}), \tag{18}$$

where $\lambda \sim \mathcal{U}(0, 1)$. Here, $f_c^{(l)}$ represents a specific batch of feature maps at a layer $l$ of the encoder $E_c$, and $\tilde{f}_c^{(l)}$ is derived by shuffling $f_c^{(l)}$ randomly across the batch dimension.

Following the method outlined by Huang & Belongie (2017), we then reconstruct the new feature maps using the perturbed feature statistics:

$$f_{new}^{(l)} = \sigma_{new}\frac{f_c^{(l)} - \mu(f_c^{(l)})}{\sigma(f_c^{(l)})} + \mu_{new}. \tag{19}$$

The perturbed feature maps are used to drive content-focused learning, thereby aiding in learning domain-agnostic embeddings.

### 3.2.3 Domain-focused Learning

Recognizing the strong correlation between image style and its domain, we prioritize extracting style features over traditional low-level features in the initial layers of the domain-specific encoder $E_d$. This approach aims to enhance the learning of domain-specific embeddings. To achieve this, we employ two widely used methods for style feature extraction.

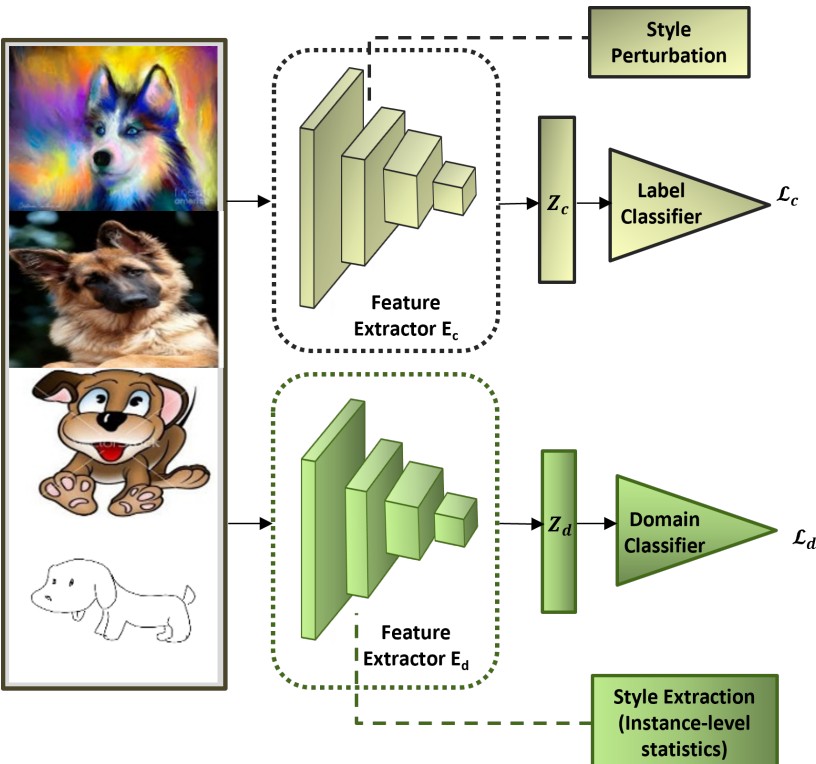

Figure 2: DETSI: Disentangled Embedding Through Style Information, leveraging instance-level feature statistics for style feature extraction to enable domain-focused learning.

**Instance-Level Feature Statistics (Mean and Variance)** We compute instance-level feature statistics, such as the mean and variance of feature activations, to represent an image's style. To capture style information, we utilize these instance-level statistics and modify the current feature maps following the methodology introduced by Huang & Belongie (2017):

$$f_{sty}^{(l)} = \mu_{sty} + \sigma_{sty} \cdot f_d^{(l)}, \tag{20}$$

where $f_d^{(l)}$ represents a specific batch of feature maps at a layer $l$ of the encoder $E_d$, and $\mu_{sty}$ and $\sigma_{sty}$ denote the style characteristics of the feature maps $f_d^{(l)}$. Instead of directly using $f_d^{(l)}$, the domain-specific embedding $Z_d$ is computed using $f_{sty}^{(l)}$.

This approach enhances domain-specific learning and strengthens the disentanglement of domain-specific and domain-invariant components. By improving this separation, we enhance the model's ability to generalize to unseen domains. Fig. 2 illustrates the DETSI framework, which leverages instance-level feature statistics for style extraction to enable domain-focused learning.

**Gram Matrix-Based Approach**  This work also considers another popular method introduced by Gatys et al. (2016) to capture the style using Gram matrices, which compute the correlations between feature channels in a CNN. The Gram matrix encodes second-order statistics (feature correlations), effectively capturing texture and style information. The Gram matrix is formally defined as:

$$G^{(l)} = f_d^{(l)} \cdot f_d^{(l)^T}, \tag{21}$$

where $f_d^{(l)}$ represents a specific batch of feature maps at a layer $l$ of the encoder $E_d$. To extract style information, Gram matrices are computed from the early layers of the domain-specific encoder $E_d$. Adaptive average pooling is applied to the Gram matrices to reduce complexity while retaining critical information, producing resized feature vectors denoted $f_{sty}^{(l)}$. These vectors are combined and passed through fully connected layers to learn the domain-specific embedding $Z_d$.

The DETSI framework leverages Gram matrices for style feature extraction to enable domain-focused learning, as illustrated in Fig. 3.

The performance analysis of both methods for style extraction is detailed in Section 5. Details about the network structure are provided in the following section.

---

**Algorithm 2** Training procedure for DETSI

---

1: **Input:** $K$ domains data samples; Encoders $E_c$, $E_d$; Classifiers $C$, $\tilde{C}$
2: **Output:** Optimized Encoder $E_c$ and Classifier $C$
3: **for** $i = 1$: epochs **do**
4:     Sample a mini-batch $(x_i^{(k)}, y_i^{(k)})_{i=1}^{B} \in S^k$
5:     $f_c^{(l)} = E_c^l(x_i^k)$
6:     $f_d^{(l)} = E_d^l(x_i^k)$
7:     $\tilde{f}_c^{(l)} = \text{Shuffle } (f_c^{(l)})$
8:     $f_{new}^{(l)} = \text{StylePerturbation } (f_c^{(l)}, \tilde{f}_c^{(l)})$
9:     $f_{sty}^{(l)} = \text{StyleExtraction } (f_d^{(l)})$
10:     Compute the objective function.
11:     $\mathcal{L}_{dI} = \mathcal{L}_c + \lambda_1 \mathcal{L}_d$
12:     Update $E_c, E_d, C, \tilde{C}$
13: **end for**
14: **Return** Optimized Encoder $E_c$ and Classifier $C$

---

## 4 Experimental Setup and Evaluation

This section details the experimental setup, including the datasets, implementation specifics, and network architectures, to evaluate the proposed methods for DG. We utilize four widely recognized benchmarks, PACS, VLCS, Office-Home, and TerraIncognita, each designed to test the robustness of DG models across diverse domains and styles. The implementation details describe the training protocols, hyperparameter configurations, and data preprocessing techniques, ensuring reproducibility.

Finally, we outline the architecture of the feature extractors, predictors, and auxiliary networks used in our experiments, emphasizing their roles in learning disentangled embeddings and leveraging style and mutual

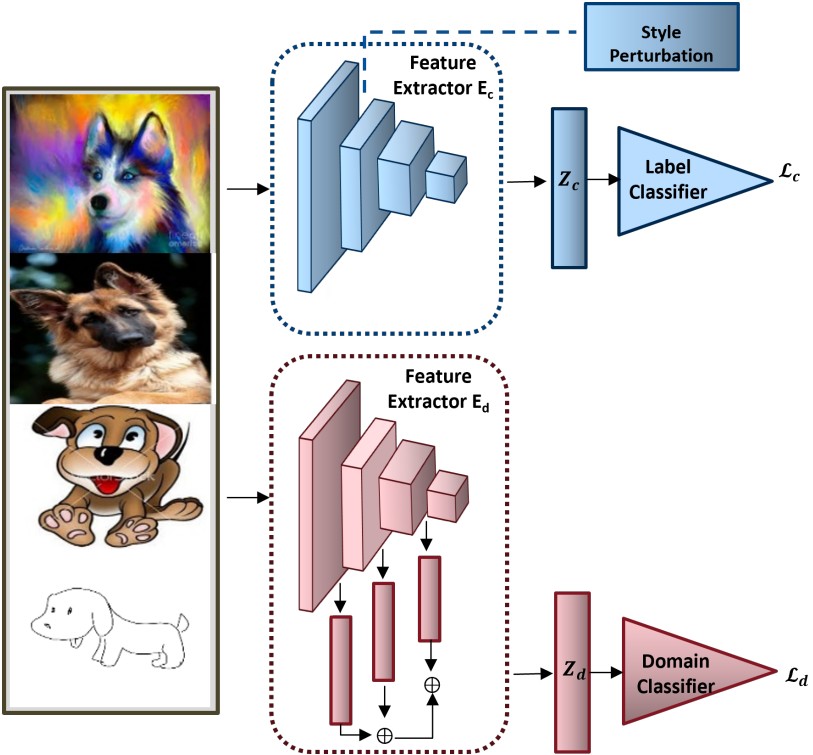

Figure 3: DETSI: Disentangled Embedding Through Style Information, utilizing Gram Matrix for style feature extraction to enable domain-focused learning.

information for improved generalization. These components collectively validate the effectiveness of our framework in addressing the challenges posed by DG.

### 4.1 Datasets

**PACS** PACS is a benchmark for object recognition in DG tasks. It includes four domains named Photo, Art-painting, Cartoon, and Sketch, each characterized by significant variations in image styles. The dataset contains 9,991 images across seven classes: dog, elephant, giraffe, guitar, horse, house, and person. This study uses the official training-validation split.

**VLCS** VLCS is another object recognition dataset comprising 10,729 images in five categories. It includes four domains: VOC 2007 (Pascal), LabelMe, Caltech, and Sun. The training and validation split follows the methodology described in Li et al. (2018b).

**Office-Home** Office-Home is designed for DG and features 15,500 images spanning 65 categories. It includes four domains: Art, Clipart, Product, and Real-world with variations in viewpoints and image styles. The training and validation split adheres to the approach outlined in Xu et al. (2021).

**TerraIncognita** TerraIncognita is a wildlife image dataset designed for domain generalization, containing 24,788 images in 10 animal categories. The data set includes four domains corresponding to camera trap locations, L38, L43, L100, and L46. These domains exhibit significant visual variability due to differences in background, lighting, and environment. Following the protocol in Cha et al. (2022), we use 80% of the data from the source domain for training and reserve the remaining 20% for validation.

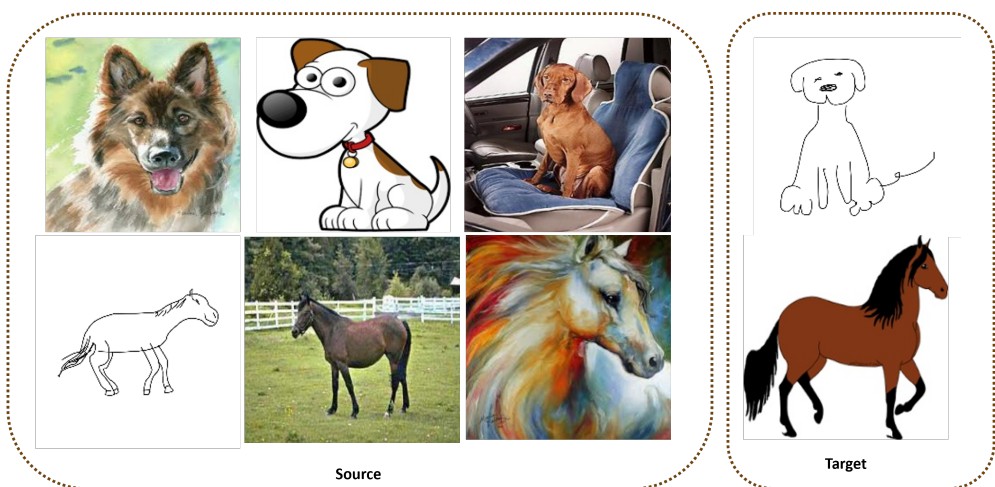

Figure 4: Training Vs. Inference Samples

## 4.2 Implementation Details

We follow the standard approach for domain generalization (DG) illustrated in Fig. 4. In this setup, one domain is designated as the target domain for testing, while the training and validation splits of the remaining domains are used to train the model and select the best-performing configuration, respectively.

The model is trained using mini-batch stochastic gradient descent (SGD) with a batch size of 32 for the PACS, Office-Home, VLCS, and and TerraIncognita datasets. The training process spans 50 epochs, with a weight decay of $5e^{-4}$ and an initial learning rate of 0.001. The learning rate is reduced by a factor of 0.1 after every 40 epochs. The weighting coefficients $\lambda_1$ and $\lambda_2$ are set to 1. Standard data augmentation techniques are applied, including color jittering, horizontal flipping, and random resized cropping. All input images are resized to 224×224 to ensure consistency during training.

## 4.3 Structure of Networks

**Feature Extractors** We utilize pre-trained ResNet50 He et al. (2016) as the feature encoders $E_c$ and $E_d$ for all datasets. The encoder $E_c$ extracts domain-agnostic features, denoted as $Z_c$, while $E_d$ extracts domain-specific features, denoted as $Z_d$.

**Predictors** The class label predictor consists of a single fully connected linear classifier $C$ with an input dimension of 2048. We use a classifier $\tilde{C}$ to predict domain labels comprising an input layer of size 2048, two hidden layers, and an output layer. The dimensions of the hidden layers are identical to the input layer, and a ReLU activation function follows each hidden layer. The output dimensions of both classifiers correspond to the number of object classes and domains in the training data.

**Network for Mutual Information Estimation** The mutual information estimator, $N_\theta$, is a neural network with two fully connected layers. The hidden layer outputs a feature vector of size 512, and the network produces a single scalar value that represents the estimated mutual information between $Z_c$ and $Z_d$.

**Network for Style Extraction in DETSI** In the DETSI framework, the style information is extracted using Gram matrices computed from the three layers of the encoder $E_d$. Experimental results indicate that the first layer captures the most significant style information compared to the subsequent layers. To optimize computational efficiency while preserving critical information, the Gram matrices initially sized 256×256 (layer 1), 512×512 (layer 2), and 1024×1024 (layer 3) are resized to a uniform dimension of 64×64 using adaptive average pooling. The resized Gram matrices are flattened and passed through a fully connected

network with ReLU activation. The resulting features are fed into the domain classifier $\tilde{C}$, ensuring a balance between computational efficiency and effective use of style information for disentanglement.

## 4.4 Results and Discussion

We employ Empirical Risk Minimization (ERM) Vapnik (2013) as a baseline, which trains a single model across all source domains by minimizing the average classification loss. In our experiments, we train the ERM model using labeled data from all available source domains and evaluate it on the unseen target domain without access to target supervision. We use ResNet-50 as the embedding encoder for the PACS, OfficeHome, VLCS, and TerraIncognita datasets. The corresponding results are presented in 1, 2, 3 and 4, respectively. The results demonstrate that the proposed frameworks, DETMI and DETSI, outperform several state-of-the-art (SOTA) approaches in terms of average accuracy across these standard benchmarks. Fig. 5 and 6 show the t-SNE Van der Maaten & Hinton (2008) visualization of domain-agnostic embeddings learned using the DETMI and DETSI frameworks, respectively.

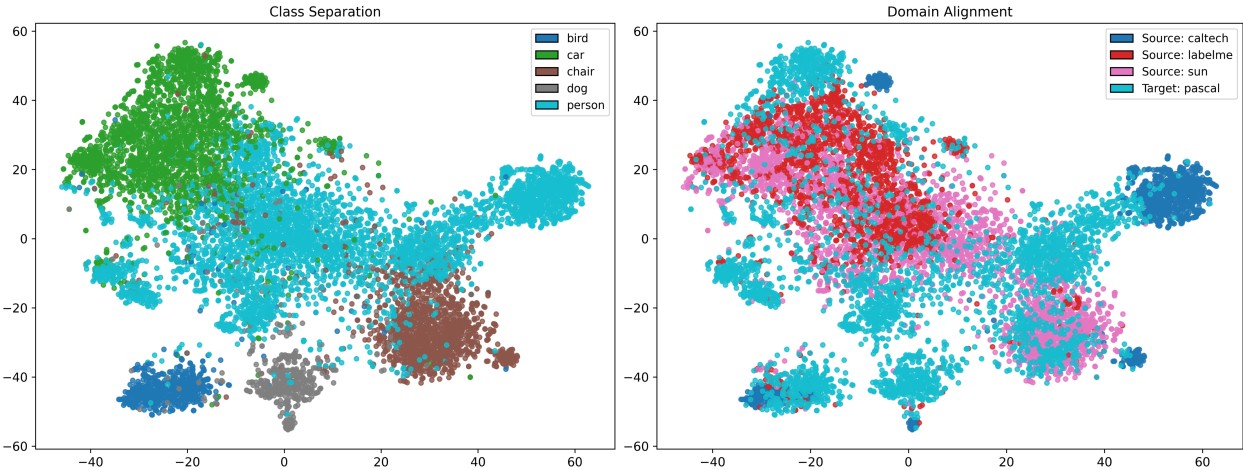

Figure 5: t-SNE plots of embeddings learned by the DETMI framework using the VLCS benchmark. (Left) Points are colored by class label to show semantic separation. (Right) Points are colored by domain to illustrate source-target alignment.

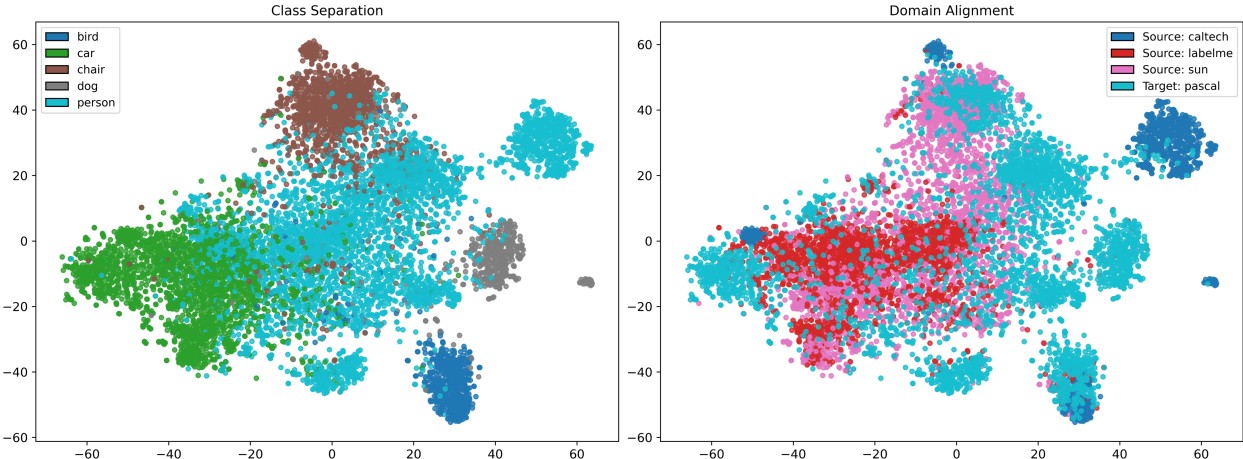

Figure 6: t-SNE plots of embeddings learned by the DETSI framework using the VLCS benchmark. (Left) Points are colored by class label to show semantic separation. (Right) Points are colored by domain to illustrate source-target alignment.

For the PACS benchmark, both approaches achieve comparable performance compared to several SOTA methods, with average accuracy improvements of +0.74%. In particular, in the sketch domain, characterized by its reliance on domain-invariant features, DETSI significantly outperforms DETMI and other SOTA methods, demonstrating its effectiveness in learning disentangled embeddings for improved performance.

For the VLCS dataset, DETMI and DETSI also demonstrate notable improvements over the SOTA methods, achieving average accuracy gains of +0.67% and +1.01%, respectively.

For the OfficeHome dataset, known for its complexity due to the large number of classes, our methods outperform several existing SOTA approaches, including POEM.

For the TerraIncognita dataset, which is considered challenging due to severe domain shifts caused by varying camera trap locations, backgrounds, lighting conditions, and class imbalance, our methods outperform existing SOTA approaches, achieving improvements of +0.94% and +0.38% in the average accuracy for DETMI and DETSI, respectively.

These results underscore the effectiveness of our methods in learning domain-agnostic embeddings, significantly enhancing the generalization capabilities of our framework to previously unseen target domains.

Table 1: Leave-one-domain-out results on PACS. The best and second-best results are bolded and underlined respectively.

| Methods | Art | Cartoon | Photo | Sketch | Avg. |
|---|---|---|---|---|---|
| | | ResNet-50 | | | |
| ERM | 86.18 | 78.58 | 97.24 | 73.55 | 83.88 |
| SagNet (CVRR 21) | 87.40 | 80.70 | 97.10 | 80.00 | 86.30 |
| MatchDG (ICML 21) | 85.61 | 82.12 | 97.94 | 78.76 | 86.11 |
| mDSDI (NeurIPS 21) | 88.10 | 81.10 | **98.40** | 79.60 | 86.80 |
| SWAD (NeurIPS 21) | 89.30 | 83.40 | 97.30 | 82.50 | 88.10 |
| DiWA (NeurIPS 22) | 90.60 | 83.40 | 98.20 | 83.80 | 89.00 |
| MIRO (ECCV 22) | - | - | - | - | 85.04 |
| POEM (AAAI 23) | - | - | - | - | 86.90 |
| CCFP (ICCV 23) | - | - | - | - | 86.60 |
| INSURE (TIP 24) | **90.20** | 85.30 | 97.90 | 83.80 | 89.30 |
| CMCL (TNNLS 25) | 87.57 | 83.60 | 96.03 | 83.73 | 87.73 |
| DETMI (Ours) | 88.76 | **85.70** | 98.08 | 82.08 | 88.65 |
| DETSI (Ours) | 90.18 | 84.85 | 98.20 | **86.94** | **90.04** |

## 5 Ablation Studies

This section presents ablation studies conducted to evaluate the DETMI and DETSI frameworks. The studies analyze key components and design choices within the frameworks to assess their impact on performance and generalization.

The first study evaluates the impact of disentangling domain-specific and domain-invariant characteristics by leveraging domain-specific knowledge and minimizing mutual information between domain-specific ($Z_d$) and domain-invariant ($Z_c$) representations. The goal is to assess how this disentanglement contributes to improved generalization performance of DETMI on unseen domains.

The second study examines the residual interaction between the domain-specific ($Z_d$) and domain-invariant ($Z_c$) feature spaces of DETSI by incorporating a mutual information estimator. This estimator quantifies any remaining mutual information between the two embedding spaces, providing information on the effectiveness of the disentanglement process.

Table 2: Leave-one-domain-out results on VLCS. The best and second-best results are bolded and underlined respectively.

| Methods | Caltech | Labelme | Pascal | Sun | Avg. |
|---|---|---|---|---|---|
| ResNet-50 | | | | | |
| ERM | 98.58 | 64.53 | 75.23 | 71.51 | 77.46 |
| SagNet (CVRR 21) | 97.90 | 64.50 | 77.50 | 71.40 | 77.80 |
| mDSDI (NeurIPS 21) | 97.60 | 66.40 | 77.80 | 74.00 | 79.00 |
| SWAD (NeurIPS 21) | 98.80 | 63.30 | 79.20 | 75.30 | 79.10 |
| DiWA (NeurIPS 22) | 98.90 | 62.40 | 73.90 | 78.90 | 78.52 |
| MIRO (ECCV 22) | - | - | - | - | 79.00 |
| POEM (AAAI 23) | - | - | - | - | 79.80 |
| CCFP (ICCV 23) | - | - | - | - | 79.20 |
| INSURE (TIP 24) | 98.80 | 63.80 | **81.04** | 72.20 | 79.01 |
| DETMI (Ours) | 98.65 | 68.22 | 78.70 | 76.32 | 80.47 |
| DETSI (Ours) | **99.15** | **68.67** | 78.34 | **77.08** | **80.81** |

Table 3: Leave-one-domain-out results on OfficeHome. The best and second-best results are bolded and underlined respectively.

| Methods | Art | Clipart | Product | Real | Avg. |
|---|---|---|---|---|---|
| ResNet-50 | | | | | |
| ERM | 67.30 | 55.80 | 78.98 | 79.06 | 70.28 |
| SagNet (CVRR 21) | 63.40 | 54.80 | 75.80 | 78.30 | 68.10 |
| mDSDI (NeurIPS 21) | 68.40 | 52.50 | 76.20 | 80.60 | 69.42 |
| DiWA (NeurIPS 22) | **69.20** | 59.00 | **80.60** | **82.20** | **72.75** |
| IIB (AAAI 22) | - | - | - | - | 68.60 |
| MIRO (ECCV 22) | - | - | - | - | 70.90 |
| POEM (AAAI 23) | - | - | - | - | 68.20 |
| CCFP (ICCV 23) | - | - | - | - | 68.90 |
| CMCL (TNNLS 25) | 67.22 | 57.88 | 78.47 | 79.79 | 70.84 |
| DETMI (Ours) | 68.58 | 58.28 | 79.22 | 81.27 | 71.83 |
| DETSI (Ours) | 67.30 | **60.29** | 78.30 | 80.03 | 71.48 |

Table 4: Leave-one-domain-out results on TerraIncognita. The best and second-best results are bolded and underlined respectively.

| Methods | L100 | L38 | L43 | L46 | Avg. |
|---|---|---|---|---|---|
| ResNet-50 | | | | | |
| ERM | 55.93 | 44.56 | 54.02 | 38.66 | 48.29 |
| SagNet (CVRR 21) | 53.00 | 43.00 | 57.90 | 40.40 | 48.60 |
| mDSDI (NeurIPS 21) | 53.20 | 43.30 | 56.70 | 39.20 | 48.10 |
| SWAD (NeurIPS 21) | 55.40 | 44.90 | 59.70 | 39.90 | 50.00 |
| DiWA (NeurIPS 22) | 57.20 | 50.10 | 60.30 | 39.80 | 51.85 |
| IIB (AAAI 22) | - | - | - | - | 47.20 |
| MIRO (ECCV 22) | - | - | - | - | 50.40 |
| POEM (AAAI 23) | - | - | - | - | 50.10 |
| CCFP (ICCV 23) | - | - | - | - | 49.00 |
| INSURE (TIP 24) | 58.80 | 46.40 | **61.70** | 45.50 | 53.10 |
| DETMI (Ours) | 60.13 | **51.53** | 57.98 | **46.50** | **54.04** |
| DETSI (Ours) | **62.24** | 50.99 | 56.41 | 44.29 | 53.48 |

The third study evaluates the impact of different style extraction techniques in DETSI to learn domain-specific embeddings ($Z_d$). It compares the performance of Gram matrices and instance-level feature statistics, with style perturbation applied solely through instance-level feature statistics across various layers.

The fourth study investigates the effect of style perturbation in the task-specific encoder and style extraction in the domain-specific encoder across different layers. The findings highlight the critical role of early encoder layers in achieving effective disentanglement.

Finally, the last study explores the independent contributions of style perturbation for task-specific embeddings, domain-specfic embeddings, and style extraction for domain-specific embeddings. This analysis underscores their complementary roles in facilitating robust disentanglement and enhancing generalization across domains.

## 5.1 Impact of Domain-Specific Knowledge and Mutual Information Minimization in DETMI

To evaluate the contribution of each component in the DETMI framework, we conduct a study that focuses on the role of domain-specific knowledge controlled through $\lambda_1$ and mutual information between domain-specific ($Z_d$) and domain-invariant ($Z_c$) representations controlled through $\lambda_2$ in 12. Our results, presented in Table 5, demonstrate that leveraging domain-specific information plays a critical role in promoting the disentanglement of domain-invariant and domain-specific representations. This structural separation enables the model to more effectively isolate invariant features that generalize across domains. Furthermore, incorporating mutual information minimization between the two feature spaces further strengthens this disentanglement by suppressing higher-order dependencies, thereby reducing redundancy. This encourages learning of more robust domain-invariant features, which ultimately leads to improved generalization performance on unseen domains.

Table 5: Impact of Domain-Specific Knowledge and Mutual Information Minimization

| Methods | Art | Cartoon | Photo | Sketch | Avg. |
|---|---|---|---|---|---|
| ResNet-50 | | | | | |
| ERM | 86.18 | 78.58 | 97.24 | 73.55 | 83.88 |
| ($\lambda_1$=1,$\lambda_2$=0) | 87.93 | 82.59 | 97.72 | 79.48 | 86.93 |
| ($\lambda_1$=1,$\lambda_2$=1) | 88.76 | 85.70 | 98.08 | 82.08 | 88.65 |

## 5.2 Validating Implicit Disentanglement in DETSI

We integrated a mutual information estimator into the DETSI framework to evaluate the residual interaction between domain-specific ($Z_d$) and domain-invariant ($Z_c$) feature spaces. This integration enabled us to quantify any remaining mutual information between the two embedding spaces. The results, summarized in Table 6, indicate that there is no significant improvement in performance with the addition of mutual information estimation. These findings confirm that the inherent design of DETSI effectively achieves robust disentanglement without the need for additional mutual information measures.

Table 6: Validating Implicit Disentanglement in DETSI using the PACS Dataset.

| Methods | Art | Cartoon | Photo | Sketch | Avg. |
|---|---|---|---|---|---|
| ResNet-50 | | | | | |
| DETSI | 90.18 | 84.85 | 98.20 | 86.94 | 90.04 |
| DETSI with MI | 89.79 | 84.72 | 98.32 | 86.43 | 89.82 |

### 5.3 Impact of Style Extraction Techniques on Learning Domain-Specific Embeddings in DETSI

This subsection presents the experimental results that evaluate the impact of different style extraction techniques on the learning of domain-specific embeddings ($Z_d$). As shown in Table 7, Gram matrix-based style extraction consistently outperforms methods based on instance-level feature statistics.

The superior performance of Gram matrices stems from their ability to encode second-order statistics, which capture correlations between feature maps across the entire image. These correlations effectively represent texture patterns, color distributions, and structural styles, all of which are critical components of an image's overall style. In contrast, instance-level feature statistics primarily focus on low-level details, such as brightness and contrast, but fail to capture global style attributes, including texture patterns, artistic brushstrokes, and structural relationships. These limitations make instance-level feature statistics less effective for representing domain-specific style information.

Table 7: Impact of Style Extraction Techniques on Learning Domain-Specific Embeddings $Z_d$ on the PACS Dataset.

| Methods | Art | Cartoon | Photo | Sketch | Avg. |
|---|---|---|---|---|---|
| ResNet-50 | | | | | |
| Instance-level Statistics | 86.69 | 84.25 | 98.08 | 85.72 | 89.43 |
| Gram Matrix | 90.18 | 84.85 | 98.20 | 86.94 | 90.04 |

### 5.4 Effect of Style Perturbation and Extraction Across Layers in DETSI

This study examines the impact of style extraction and perturbation across the early layers of encoders in the DETSI framework, using instance-level feature statistics to learn domain-specific embeddings.

The results, summarized in Table 8, reveal that the application of style extraction and perturbation within the first three layers of the domain-specific encoders achieves superior performance compared to other layer combinations. These findings align with the established understanding that early encoder layers primarily capture low-level features, such as textures and patterns, which are closely associated with style.

Interestingly, the fourth layer, which primarily encodes high-level semantic features rather than style attributes, was excluded from this analysis. This exclusion underscores the critical role of early encoder layers in style manipulation to enhance the disentanglement of domain-specific and domain-agnostic embeddings in the DETSI framework.

Table 8: Evaluation of Style Feature Extraction and Perturbation Across Different Layers of DETSI on the PACS Dataset

| Methods | Art | Cartoon | Photo | Sketch | Avg. |
|---|---|---|---|---|---|
| ResNet-50 | | | | | |
| $E_c(L1), E_d(L1)$ | 89.79 | 84.51 | 98.08 | 82.92 | 88.82 |
| $E_c(L12), E_d(L12)$ | 89.59 | 84.00 | 98.32 | 84.72 | 89.15 |
| $E_c(L123), E_d(L123)$ | 89.69 | 84.25 | 98.08 | 85.72 | 89.43 |

### 5.5 Evaluation of DETSI Components on Domain Generalization Benchmarks

This ablation study examines the impact of style perturbation on task-specific embeddings, domain-specific embeddings, and style extraction on domain-specific embeddings within the DETSI framework. As shown in Table 9, each component contributes to the performance gains, and the combined application yields significant improvement. These results highlight the roles of style perturbation and style extraction to enhance feature disentanglement. Ultimately, the framework increases generalization across unseen domains.

Table 9: Evaluation of DETSI Components on Domain Generalization Benchmarks

| Methods | PACS | VLCS | OfficeHome |
|---|---|---|---|
| ERM | 83.88 | 77.46 | 70.28 |
| +Style Pert. in $E_c$ [†] | 85.20 | 77.90 | 60.40 |
| +Style Pert. in $E_c$ +$E_d$ | 89.58 | 79.76 | 70.71 |
| +Style Pert. in $E_c$ + Style Extr. in $E_d$ | 90.04 | 80.81 | 71.48 |

[†]Result adopted from Zhou et al. (2024) due to matching configuration.

## 6 Conclusion

This paper addresses the challenge of domain shift in Domain Generalization (DG) by introducing two novel frameworks: Disentangled Embedding through Mutual Information (DETMI) and Disentangled Embedding through Style Information (DETSI). These frameworks effectively leverage domain-specific information to disentangle the latent feature space into domain-specific and domain-invariant components, enabling the extraction of class-relevant features and enhancing generalization to unseen domains.

DETMI utilizes a mutual information estimator to promote feature disentanglement, while DETSI achieves disentanglement through leveraging style information. Both frameworks perform better than state-of-the-art DG techniques, promoting domain invariance and improving generalization. Furthermore, DETSI achieves comparable results with reduced complexity, making it a practical and efficient solution for scenarios with limited computational resources.

The proposed frameworks advance the state-of-the-art in DG and highlight the critical role of leveraging domain-specific information alongside domain-invariant features. These findings underscore the potential of disentanglement-based approaches to effectively address domain shift and provide a foundation for developing more efficient and robust DG methods in the future.

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
