# OpenReview forum: "Disentangled Embedding through Style and Mutual Information for Domain Generalization"
_TMLR — Accepted by TMLR_

### Review · Reviewer_TXmC · 2025-04-23

**Summary Of Contributions:**

This paper presents two frameworks, namely Disentangled Embedding through Mutual Information (DETMI) and Disentangled Embedding through Style Information (DETSI), to separate domain-invariant and domain-specific features. DETMI minimizes the mutual information between learned embeddings, while DETSI extracts style statistics (instance-level feature statistics & Gram) and introduces style perturbation. Both approaches deliver strong results on PACS, Office-Home, and VLCS.

**Audience:**

Yes

**Claims And Evidence:**

Yes

**Requested Changes:**

Please see weakness part.

Additionally, I feel like the framework figures in the paper are quite similar to each other, which can be somewhat distracting in their current form. It may be more effective to present them side by side for a clearer comparison and overall clarity.

**Strengths And Weaknesses:**

**Strength:**

The paper is quite easy to follow, with a comprehensive survey of related work.

The experimental design is clear, and it demonstrates the effectiveness of the proposed methods on the three benchmarks.


**Weakness:**

-   Incremental Novelty: the dual-encoder setup with a domain classifier and a label classifier is common for DG. DG by mutual information is also explored (for example MIRO). Adding a mutual-information term to encourage independence between learned embeddings, while shows promising results in the paper, is a relatively straightforward extension rather than a paradigm shift. Likewise, instance normalization and Gram are both explored.

-   Limited Evaluation Scope: while PACS, Office-Home, and VLCS used to be popular DG benchmarks, they feature only a few domains with pronounced style differences. It remains unclear how the proposed methods scale to more challenging benchmarks such as DomainNet, or to real-world distribution shifts such as  WILDS-FMoW or TerraIncognita.

---

> ### Author Response · Authors · 2025-05-15
> **Response with thanks for your valuable feedbac**
>
> First of all, we thank the reviewer for the insightful feedback.
> >Incremental Novelty: the dual-encoder setup with a domain classifier and a label classifier is common for DG. DG by mutual information is also explored (for example MIRO). Adding a mutual-information term to encourage independence between learned embeddings, while shows promising results in the paper, is a relatively straightforward extension rather than a paradigm shift. Likewise, instance normalization and Gram are both explored.
>
> We acknowledge that components such as dual-encoder architectures, mutual information (MI) regularization and instance normalization have been individually explored in prior domain generalization (DG) work (MIRO, POEM). However, our contribution lies in how we reformulate and integrate these components into two unified and complementary frameworks—DETMI and DETSI—that explicitly leverage domain-specific cues to disentangle domain-invariant representations.
> MIRO enhances DG by preserving generalizable semantic features learned by large-scale pre-trained models (e.g., CLIP, SWAG). It does so by maximizing mutual information between task-specific features and a frozen oracle, thereby encouraging the retention of transferable representations during fine-tuning. In contrast, DETMI builds on a pre-trained backbone but follows a fundamentally different objective: it introduces domain-specific and domain-invariant encoders and applies mutual information minimization between their outputs. This encourages deeper statistical independence and explicitly suppresses domain-specific dependencies, resulting in more robust domain-agnostic representations. Furthermore, DETSI uses style perturbation and extraction motivated by findings from style transfer literature to explicitly model and isolate style as a domain-specific factor, guiding the encoder to ignore superficial variations and focus on semantic content.
> While the individual components used in our frameworks may not be entirely novel, their specific formulation, purposeful integration, and application toward structured disentanglement in domain generalization constitute the core contribution of this work. Our extensive evaluations (including TerraIncognita) demonstrate consistent performance gains over existing state-of-the-art methods, including both MIRO and POEM. We hope this clarifies the contribution of our proposed frameworks in relation to prior work.
>
> >Limited Evaluation Scope: while PACS, Office-Home, and VLCS used to be popular DG benchmarks, they feature only a few domains with pronounced style differences. It remains unclear how the proposed methods scale to more challenging benchmarks such as DomainNet, or to real-world distribution shifts such as WILDS-FMoW or TerraIncognita.
>
> In addition to PACS, VLCS, and Office-Home, we have included an evaluation on TerraIncognita, a more challenging benchmark known for its real-world domain shifts across camera trap locations. This benchmark introduces significant intra-class variability, severe domain shifts, and unstructured noise, making it well-suited for testing the robustness of domain generalization methods. The strong performance of our proposed frameworks on TerraIncognita further validates their effectiveness beyond stylized datasets.
> While we acknowledge the value of DomainNet for evaluating domain generalization, DomainNet’s large size (~0.6M images) presents practical constraints in terms of computational resources and training time. Due to these limitations, we opted not to include DomainNet in our current evaluation. We believe the current set of benchmarks offers a representative and diverse evaluation of our methods within feasible resource constraints.
>
> >Additionally, I feel like the framework figures in the paper are quite similar to each other, which can be somewhat distracting in their current form. It may be more effective to present them side by side for a clearer comparison and overall clarity.
>
> In the revised version, we have improved the visual design of the figures to enhance clarity and reduce redundancy. Specifically, we have addressed the similarity between the DETMI and DETSI diagrams by refining their visual distinctions.

---

### Review · Reviewer_6KZa · 2025-04-23

**Summary Of Contributions:**

This paper proposes a Disentangled Embedding through Mutual Information (DETMI) and a Disentangled Embedding through Style Information (DETSI) for domain generalization, both inspired by recent advances in representation disentanglement. DETMI disentangles domain-agnostic and domain-specific embeddings by employing a mutual information estimator, while DETSI achieves this by style extraction and perturbation. Experiments on the PACS, Office-Home, and VLCS datasets show that both frameworks achieve state-of-the-art DG performance.

**Audience:**

Yes

**Broader Impact Concerns:**

There is no concerns on the ethical implications.

**Claims And Evidence:**

No

**Requested Changes:**

Please address the questions in Cons and revise the manuscript accordingly.

**Strengths And Weaknesses:**

**Pros**:
+ This paper is generally well-written and easy to follow.
+ The experiments can verify some of the key parts of the proposed method, like the Gram Matrix style extraction scheme
+ Both the DETMI and DETSI achieve better or comparable performance to state-of-the-art methods.

**Cons**:

1. There are many related works to be comprehensively compared to highlight the differences between the proposed methods and the existing ones.
Firstly, the basic idea of the feature disentanglement used in this paper is not new because it has been explored in POEM (Jo & Yoon, 2023).
Secondly, mutual information has been used in domain generalization [A] and unsupervised domain adaptation [B]. Please compare the DETMI with these methods.
Thirdly, the content-focused learning in Sec. 3.2.2 is very similar to MixStyle [C]. Please clarify their differences and compare their performance.

[A] Junbum Cha et. al. Domain Generalization by Mutual-Information Regularization with Pre-trained Models. ECCV 2022

[B] Jiahong Chen et. al. Preserving domain private information via mutual information maximization. Neural Networks 2024.

[C] Kaiyang Zhou et. al. MixStyle Neural Networks for Domain Generalization and Adaptation. IJCV, 2023

2. I do not understand why this paper first proposes DETMI with mutual information and then presents DETSI which is not compatible with mutual information (evidenced by the result in Tab. 4). Please explain why these two incompatible strategies are introduced in the same paper.

3. In Ln. 3 of Alg. 1, please explain why and how the mutual information estimator $N_\theta$ is updated by maximizing Eq (5). Eq. (5) states the case when I(X, Y)=0, so it seems not an objective function.

4. Please evaluate the sensitivity of lambda_1 and lambda_2. Particularly, the results of lambda_2=0 vs. lambda_2!=0 are necessary to verify the effectiveness of the mutual information design in DETMI.

5. In Tab. 7, the style perturbation, or MixStyle, has been verified effective in previous work. The proposed style extraction only improves it by 0.46% on the small-scale dataset, PACS. Such improvement can hardly be considered as "significant" as it can even be caused by different random seeds. Besides, if the style extraction is based on the Instance-level Statistics, there will be no improvement.

6. Please visualize the feature embeddings of the DETMI and DETSI.

**Minor concerns**:

7. Please prove that "by reducing higher-order correlations, DETMI ensures statistical independence". It is not easy to understand that reducing higher-order correlations can "ensure" statistical independence because the mutual information can hardly be 0.

8. To facilitate comparison, please include the baseline results in Tab. 4.

---

> ### Author Response · Authors · 2025-05-15
> **Response with thanks for your valuable feedback**
>
> First of all, we really appriciate the precious feedback.
> > There are many related works .....
>
> 1. Comparison with [A, B] regarding use of Mutual information
>
> MIRO [A] enhances domain generalization by preserving generalizable semantic features learned by large-scale pre-trained models. It maximizes mutual information between the learned features and those of a frozen pre-trained oracle, thereby retaining transferable knowledge during fine-tuning.
> In contrast, DETMI is built upon a pre-trained backbone but focuses on explicit disentanglement. DETMI introduces parallel domain-specific and domain-invariant encoders and minimizes mutual information between their outputs. This enhances statistical independence and suppresses domain-specific noise in the learned representations. Our experimental results demonstrate consistent performance gains over MIRO. Also, mutual information has also been applied in domain adaptation (e.g., Chen et al., [B]); the goal differs fundamentally from domain generalization. In domain adaptation, the target domain is available during training, and MI is typically maximized to retain shared information between the source and target domains. In contrast, domain generalization operates without access to the target domain, making mutual information minimization across feature components a more suitable strategy for learning transferable, domain-invariant representations.
>
> 3. Distinction from MixStyle[3]:
>
> While MixStyle applies mixup on domain-invariant features to improve generalization, it does not explicitly model domain-specific information. As a result, its performance** saturates at 85.2% on PACS, 77.9% on VLCS, 60.4% on OfficeHome, and 44.0% on TerraIncognita. In contrast, our DETSI framework integrates a dedicated domain-specific encoder and classifier to model domain-specific signals explicitly while learning domain-invariant embeddings. Moreover, DETSI incorporates a style extraction and purturbation mechanism based on the hypothesis that style is correlated with domain, which improves disentanglement between domain-specific style and semantic content. DETSI outperforms Mixstyle by an excellent margin on the four DG benchmarks.
>
> > I do not understand ...why these two incompatible strategies are introduced in the same paper.
>
> While DETMI and DETSI employ different mechanisms, both frameworks are fundamentally built upon the disentanglement principle and contribute complementary perspectives to the disentanglement-based approach for domain generalization. We believe exploring these strategies in the same paper strengthens the contribution and provides a broader understanding of how disentanglement can be achieved effectively through different means.
>
> >In Ln. 3 of Alg. 1,....
>
> The confusion arose due to a typographical error in the equation reference which has now been corrected in the revised version. We apologize for the oversight and appreciate the reviewer’s careful reading.
>
> >Please evaluate the sensitivity ...
>
> In the revised version, we have added an ablation study that analyzes the sensitivity of the loss coefficients lambda_1 and lambda_2.  We believe this analysis (Table 5)  further supports the design choice behind our proposed framework.
>
> >In Tab. 7, the style perturbation, or MixStyle, has been verified effective in previous work. The proposed style extraction only improves it by 0.46% ...
>
> As addresed in earlier response, MixStyle applies feature-space mixup to encourage generalization, it does not explicitly model domain-specific information. The 0.46% gain on PACS reflects the incremental contribution of the style extraction component within DETSI—not the overall gain over MixStyle. The total performance improvement of DETSI over MixStyle is much more substantial.
>
> >Please visualize the feature....
>
> In the revised version of the manuscript, we have included t-SNE visualizations of the feature embeddings learned by both DETMI and DETSI which provide qualitative insight into the separation of domain-invariant features and further support the effectiveness of our frameworks.
>
> >Please prove that "by reducing higher-order correlations...
>
> We agree with the observation that mutual information rarely reaches zero in practice, and thus, the original phrasing suggesting it "ensures" statistical independence was too strong. In the revised version of the manuscript, we have reworded the statement to reflect that minimizing mutual information serves to reduce higher-order dependencies, which in turn enhances the disentanglement between domain-specific and domain-agnostic embeddings. We appreciate the reviewer’s attention to this important clarification.
>
> >To facilitate comparison, please include the baseline results in Tab. 4.
>
> In the revised version, we have included the baseline results in the updated Table 6 (previously Table 4) to facilitate clearer comparison.
>
> **resutls from POEM where the same encoder architecture (ResNet-50) was used, ensuring a fair comparison.

---

> > ### Comment · Reviewer_6KZa · 2025-05-20
> > **Further discussion**
> >
> > Thanks for the response which has addressed some of my concerns. However, some concerns persist:
> >
> > For Q1, the MixStyle “saturates at 85.2% on PACS”. However, the baseline of the proposed method, which is ERMVapnik (2013) as in Tab. 7 (i.e., Tab. 9 in the revision), has already obtained 85.50% on PACS. This implies that the comparison with MixStyle may not be fair because the proposed method benefits from a more powerful baseline which has already beaten MixStyle.
> >
> > For Q5, please note that this concern is about the effectiveness of the Style Extraction in Tab. 7 (which is Tab. 9 in the revision). This table can verify the effectiveness of Style Perturbation ($E_c$), which shows no fundamental differences from MixStyle. It implies that Style Perturbation/MixStyle can be effective in improving a powerful baseline ERMVapnik (2013). However, the main contribution of Style Extraction only shows incremental improvement (0.46% on the small-scale PACS) which may be caused by different random seeds. Specifically, adding the Style Extraction to Style Perturbation ($E_c$), which has already obtained an average accuracy of 89.58% on the small-scale PACS, we obtain DETSI, which achieves 90.04%. If we use Instance-level Statistics for the Style Extraction, achieving 89.43% as in Tab. 5 (i.e., Tab. 7 in the revision), it can decrease the performance of Style Perturbation ($E_c$) from 89.58% to 89.43%. In this case, the effectiveness of the Style Extraction cannot be justified.

---

> > > ### Author Response · Authors · 2025-05-21
> > > **Follow-up on Comments and Clarifications**
> > >
> > > >For Q1, the MixStyle “saturates at 85.2% on PACS”. However, the baseline of the proposed method, which is ERMVapnik (2013) as in Tab. 7 (i.e., Tab. 9 in the revision), has already obtained 85.50% on PACS. This implies that the comparison with MixStyle may not be fair because the proposed method benefits from a more powerful baseline which has already beaten MixStyle.
> > >
> > > We acknowledge the reviewer’s concern regarding the baseline strength used in our comparison. The ERM baseline score of 85.50% on the PACS dataset, originally reported by Yu et al*. (2022), is based on a standard implementation of ERM using a ResNet-50 backbone. For context, in the MixStyle** paper, the reported performance using ResNet-18 as the backbone is 83.70%, and when upgraded to ResNet-50, MixStyle achieves 85.20%. Importantly, the MixStyle paper itself notes that as the backbone model becomes larger and more expressive (e.g., from ResNet-18 to ResNet-50), MixStyle’s advantage over ERM diminishes, and in some benchmarks like OfficeHome, it even underperforms ERM by a notable margin.
> > > Across PACS, VLCS, and OfficeHome, MixStyle** using ResNet-50 reports 85.20%, 77.90%, and 66.40%, respectively,
> > > while the corresponding ERM baseline yields 84.20%, 77.30%, and 67.60%. In contrast, our proposed method DETSI consistently outperforms both the ERM and the MixStyle across all evaluated benchmarks. Importantly, the performance improvement is not solely due to style perturbation, but also due to the explicit disentanglement and exploitation of domain-specific embeddings learned through a dedicated domain-specific encoder.
> > >
> > > *Yu, Xi, et al. "INSURE: IEEE Transactions on Image Processing (2024).
> > >
> > > **Zhou, Kaiyang, et al. "Mixstyle neural networks for domain generalization and adaptation." International Journal of Computer Vision 132.3 (2024): 822-836.
> > >
> > > >For Q5,
> > >
> > > We acknowledge that the direct performance gain from the style extraction process within the domain-specific encoder on the PACS dataset (0.46%) may appear modest. However, as previously stated, style extraction is designed as a complementary component within the domain-specific encoder of the broader DETSI framework. Its primary role is to facilitate the learning of disentangled, domain-specific embeddings, which are subsequently leveraged to enhance the generalization capability of the learned features. While the standalone improvement of this specific component on PACS is incremental, the overall DETSI framework consistently outperforms state-of-the-art domain generalization methods, including MixStyle, across all evaluated benchmarks. This demonstrates the robustness and effectiveness of the proposed framework. Due to current time constraints, we were not able to conduct a broader empirical study of the style extraction module across additional DG benchmarks; however, we plan to explore its impact more comprehensively in future work.

---

> > > > ### Comment · Reviewer_6KZa · 2025-05-21
> > > > **Further discussion on the Style Extraction**
> > > >
> > > > Thanks for further clarification. However, the explanation cannot address my concern about the effectiveness of the Style Extraction. “The overall DETSI framework consistently outperforms state-of-the-art domain generalization methods”. This can be owed to the extra encoder $E_d$ for the feature ensemble and the Style Perturbation in $E_c$. However, the Style Perturbation is nothing different from MixStyle and using another encoder for ensemble is a well-known strategy for better performance. Thus, the contribution part is the Style Extraction, of which the importance should be verified by experiments. Please note that the effectiveness of using an extra encoder $E_d$ is not a concern. Instead, the effectiveness of the Style Extraction is a major concern. As such, please do not mix up these two.
> > > >
> > > > Regarding “its primary role is to facilitate the learning of disentangled, domain-specific embeddings, which are subsequently leveraged to enhance the generalization capability of the learned features”, this statement is not supported by the experimental results, as the learned features only achieve a negligible performance gain over the Style Perturbation ($E_c$). In this case, the effectiveness of enhancing “the generalization capability of the learned features” is debatable.
> > > >
> > > > I summarize my concern again as below. To me, the ablation study should be as follows: (1) the baseline of ERM, obtaining 85.50% in Tab. 9; (2) ERM+Style Perturbation, achieving 89.58% in Tab. 9; (3) ERM+Style Perturbation+An extra encoder $E_d$ with domain classifier (without Style Extraction), which is missing; (4) ERM+Style Perturbation+An extra encoder $E_d$ with domain classifier+ Style Extraction (90.04% in Tab. 9). Note that (4) improves (2) by only 0.46% which different random seeds on the small-scale PACS can cause. In addition, we are unsure whether the 0.46% comes from the extra encoder $E_d$ with the domain classifier or Style Extraction because the result of (3) is missing.

---

> > > > > ### Author Response · Authors · 2025-05-21
> > > > > **further clarification of DETSI components**
> > > > >
> > > > > Thank you for summarizing your concern so clearly. We really appreciate it.
> > > > >
> > > > > To clarify, style perturbation in our work refers to ERM + Style Perturbation + Domain-Specific Encoder (without the style extraction process), which achieves 89.58%. Style Extraction refers to the full DETSI framework, i.e., ERM + Style Perturbation + Domain-Specific Encoder, with the Style Extraction process, achieving 90.04%. We acknowledge that the result for ERM + Style Perturbation only is not explicitly included. This configuration is conceptually similar to MixStyle*, which reports 85.20% using ResNet-50, compared to 84.20% for ERM, as reported in the original MixStyle paper.
> > > > >
> > > > > You are also correct in noting that the contribution of the Style Extraction process within the domain-specific encoder (0.46%) on PACS appears nominal. However, we believe its significance grows as the number of domains increases, particularly when those domains exhibit diverse style characteristics. We have already begun analyzing the impact of this component on other domain generalization benchmarks.
> > > > >
> > > > > We would also like to clarify that, although our framework employs two encoders, the domain-specific encoder ($E_d$) is not used as part of a conventional ensemble. In typical ensemble learning, multiple models are trained on the same task, and their outputs are aggregated to improve predictive performance. In contrast, our domain-agnostic encoder ($E_c$) and domain-specific encoder ($E_d$) are designed to serve distinct roles, leveraging class labels and domain labels, respectively. $E_c$ learns semantic, domain-invariant representations, while $E_d$ focuses on capturing domain-specific information. The outputs of these encoders are not combined through ensemble aggregation, but are instead jointly optimized to achieve a disentanglement of content and style. This design is intended to enhance the model’s ability to generalize effectively to unseen domains by leveraging both invariant and domain-aware features.
> > > > >
> > > > > *Zhou, Kaiyang, et al. "Mixstyle neural networks for domain generalization and adaptation." International Journal of Computer Vision 132.3 (2024): 822-836.

---

> > > > > > ### Comment · Reviewer_6KZa · 2025-05-22
> > > > > > **More comprehensive ablation study is appreciated**
> > > > > >
> > > > > > “We acknowledge that the result for ERM + Style Perturbation only is not explicitly included. This configuration is conceptually similar to MixStyle*, which reports 85.20% using ResNet-50, compared to 84.20% for ERM. It seems that these two results (85.2% and 84.2%) are not reported in Tab. 9.
> > > > > >
> > > > > > “We have already begun analyzing the impact of this component on other domain generalization benchmarks.” I appreciate this, and please update the results to verify the effectiveness of all the key designs once they are obtained. Here, I suggest following the previously mentioned four settings to complete the ablation study.

---

> > > > > > > ### Author Response · Authors · 2025-05-24
> > > > > > > **outcome from the comprehensive ablation study**
> > > > > > >
> > > > > > > We thank the reviewer for the constructive suggestion regarding the comprehensive evaluation of the style extraction component in the DETSI framework. Following the recommendation, we conducted a comprehensive ablation study across three standard domain generalization benchmarks.
> > > > > > >
> > > > > > > Specifically, we observed that incorporating the style extraction component led to consistent improvements of 0.46% on PACS, 1.05% on VLCS, and 0.77% on OfficeHome over the corresponding baselines. Given the consistent nature of these improvements across diverse datasets, we believe the gains are not attributable to random seed initialization or noise.
> > > > > > >
> > > > > > > Furthermore, we hypothesize that as the number of domains increases with higher style diversity, the contribution of the style extraction module will become more pronounced. These findings support the effectiveness of this component as a meaningful addition to the overall architecture.
> > > > > > >
> > > > > > > We have updated the manuscript to include these results

---

> > > > > > > > ### Comment · Reviewer_6KZa · 2025-05-24
> > > > > > > > **No further concerns**
> > > > > > > >
> > > > > > > > Many thanks for the update. I have no further concerns.

---

### Review · Reviewer_vwZz · 2025-05-05

**Summary Of Contributions:**

This paper presents two methods to improve image classification when training with multiple domains and testing on another unknown domain, which is normally referred as domain generalization. Both approaches are based on separating the model representation in two complementary parts, the first is domain invariant and the second is domain specific. The first approach is based on minimizing the mutual information between the two representations (domain-invariant and domain-specific) using MINE.
The second approach is based on style disentaglement, in which the style of an image is substituted with the style of another image using AdaIN, such that the model would learn to focus on content and overlook style.

**Audience:**

Yes

**Broader Impact Concerns:**

I do not see any concerns about ethical implications of the work.

**Claims And Evidence:**

No

**Requested Changes:**

\- Add missing datasets: TerraIncognita and DomainNet

\- Include methods based on ensembling

\- Improve presentation of the paper as explained in the previous section

\- Explain and motivate more concretely what are the contributions of this paper

\- Correct the statements about domain adaptation in the intro or explain your way of understanding that

**Strengths And Weaknesses:**

\+ The presentation of the paper is clean and easy to understand

\+ The proposed approaches are meaningful and compelling

\- The experimental evaluation is weak. In domain generalization is based on domainbed (Gulrajani et al.2020), which includes PACS, VLCS, office-Home, but also TerraIncognita and DomainNet. Those two are larger datasets and seem more challenging and I think it is important to validate the proposed methods on those datasets. In addition, there are methods based on ensembling (Lakshminarayanan et al. 2017 and Rame et al. 2022), which are quite competitive and not included in the evaluation.

\- Even if the presentation of the paper is simple, there are several points that need to be improved. In many equations (e.g. eq. 1 and 2), there are several variables that are not defined (\delta, \sigma). The same image is used multiple times for the different approaches. There could be a better way to present that without showing multiple times the same image, which seems a bit repetitive.

\- The introduction does not motivate much the proposed contributions. It looks more like an additional related works.

\- The proposed methods are not new (MINE, AdaIN). The novelty stems form the application of the methods to domain generalization. While, this is interesting, the contribution is limited.

\- In the introduction, the authors talk about domain generalization which tries to improve generalization. In my understanding this is misleading, as in domain adaptation the aim is not to generalize, but to adapt a model to perform well on the target dataset, without caring about the performance on other domains.

---

> ### Author Response · Authors · 2025-05-15
> **Response with thanks for your valuable feedback**
>
> First of all, we thank the reviewer for the insightful feedback.
> >The experimental evaluation is weak. In domain generalization is based on domainbed (Gulrajani et al.2020), which includes PACS, VLCS, office-Home, but also TerraIncognita and DomainNet. Those two are larger datasets and seem more challenging and I think it is important to validate the proposed methods on those datasets. In addition, there are methods based on ensembling (Lakshminarayanan et al. 2017 and Rame et al. 2022), which are quite competitive and not included in the evaluation.
>
> In addition to PACS, VLCS, and Office-Home, we have now included an evaluation on TerraIncognita (one of the two larger suggested benchmarks), which further validates the effectiveness of our frameworks against existing state-of-the-art DG methods. While we acknowledge the significance of DomainNet, its large scale (approximately 0.6 million images) requires considerable computational resources and extended training time. Due to these practical constraints, we opted not to include DomainNet in our current evaluation, but we consider it an important direction for future work. Furthermore, we have now included the ensemble-based domain generalization method by Rame et al. (2022) in both our introduction and experimental evaluation. Regarding the method by Lakshminarayanan et al. (2017), we note that it was not originally evaluated in the context of domain generalization. Therefore, we did not include it in our comparisons to maintain consistency with domain-generalization-specific baselines.
>
> >Even if the presentation of the paper is simple, there are several points that need to be improved. In many equations (e.g. eq. 1 and 2), there are several variables that are not defined (\delta, \sigma). The same image is used multiple times for the different approaches. There could be a better way to present that without showing multiple times the same image, which seems a bit repetitive.
>
> We appreciate the feedback regarding clarity and presentation. We have now defined all previously undefined variables in the revised version. Additionally, we have improved the visual presentation of the framework figures to minimize redundancy.
>
> >The introduction does not motivate much the proposed contributions. It looks more like an additional related work. The proposed methods are not new (MINE, AdaIN). The novelty stems form the application of the methods to domain generalization. While, this is interesting, the contribution is limited.
>
> We acknowledge that individual components, such as mutual information estimation (e.g., MINE) and style-related normalization techniques (e.g., AdaIN), have been previously proposed. However, the contribution of our work lies not in the novelty of isolated components but in how we strategically reformulate and integrate them within the domain generalization (DG) context to address feature disentanglement in a principled and effective manner.
>
> DETMI leverages domain-specific information and incorporates mutual information minimization to enhance the disentanglement of domain-invariant features. It encourages statistical independence, enabling the model to learn robust, transferable representations that generalize well to unseen domains. Similarly, DETSI draws from insights in style transfer (e.g., AdaIN), but differs fundamentally in its objective and design. DETSI uses a dedicated domain-specific encoder and a style extraction and perturbation mechanism to isolate domain-specific style from semantic content explicitly. Unlike generic normalization, our approach treats style as a domain-dependent factor. It uses it to guide disentanglement in a way not explored in previous DG frameworks.
>
> While we agree that the base components are not entirely novel, our targeted formulation, architectural integration, and demonstrated improvements across multiple challenging DG benchmarks contribute meaningfully to the literature. In the revised manuscript, we have clarified the scope of our contributions.
>
>
>
> > In the introduction, the authors talk about domain generalization which tries to improve generalization. In my understanding this is misleading, as in domain adaptation the aim is not to generalize, but to adapt a model to perform well on the target dataset, without caring about the performance on other domains.
>
> Upon revisiting the introduction, we agree that the statement was potentially misleading, so we have removed this part in the revised version to avoid confusion and to ensure the introduction more accurately reflects the focus of domain generalization. We appreciate the insightful comment.

---

### Decision · Action_Editor_iUFa · 2025-06-05

**Recommendation:** Accept with minor revision

**Additional Comments:**

This paper investigate the use of Mutual information and style perturbation for domain generalization. Reviewers found the paper interesting if a bit incremental (the terms have been propose don previous paper albeit not for domain generalization). Some comments and discussion about the numerical experiments (and a discussion) lead to an improved version of the paper with new results including among other a more in-depth ablation study.

All reviewers agree after discussion and new revision that the paper is interesting and the experiment are rigorous. The paper fits all requirements for publications in TMLR so I will recommend and accept. But a few remaining problems raised by the reviewers need to be addressed with minor revisions. More details below.

Minor revisions:
+ ERM (no DG but learned on all source domains concatenated) is missing form the main results tables. It is important to re-run with the used architecture/training setup because it is the real baseline (and can differ between papers). It is a relatively small experiment (with no hyperparameters)  and it is much more rigorous than reporting ERM form other papers (as done currently in the ablation where it shoud have been run there).
+ The authors did report the DiWA method but did no go all the way. It is still missing on Tab3 OfficeHome and and Tab4 TerraIncognita for no reason. Also the details of performance for each domain is provided in the appendix of the DiWa paper and should be reported.
+ The tSNE visualization is interesting but could be done more in detail. For instance by doing a TSNE of both samples from target and sources and color them by domain (to check that they are indistinguishable). In this sens one TSNE embedding can be used to check both class discrimination and Domain Generalization (also classes discrimination across domains which is the core of DG). Also the class legend is wrong with colors in the scatterplots not corresponding to the legend.

**Audience:**

Yes

**Audience Explanation:**

This is an important question and while the strategies already exist in other ML applications their use for domain generalization is quite new and well tested experimentally in the paper.

**Claims And Evidence:**

Yes

**Claims Explanation:**

All the claim made in the paper are supported by the numerical experiments in particular the new ablation studies.